# How Do the Different Types of Maternal Diabetes during Pregnancy Influence Offspring Outcomes?

**DOI:** 10.3390/nu14183870

**Published:** 2022-09-19

**Authors:** Lina Eletri, Delphine Mitanchez

**Affiliations:** 1Department of Neonatology, Centre Hospitalier du Mans, 72037 Le Mans, France; 2Department of Neonatology, Bretonneau Hospital, François Rabelais University, 37000 Tours, France; 3INSERM UMRS_938, Centre de Recherche Saint Antoine, 75012 Paris, France

**Keywords:** gestational diabetes mellitus, type 2 diabetes, type 1 diabetes, obesity, overweight, cardiovascular diseases, metabolic syndrome, neurodevelopmental disorders, autism, attention deficit hyperactive disorder

## Abstract

***Background/Aim of the study***: Exposure to maternal diabetes is considered one of the most common in utero insults that can result in an increased risk of complications later in life with a permanent effect on offspring health. In this study, we aim to assess the level of risk associated with each type of maternal diabetes on obesity, glucose intolerance, cardiovascular diseases (CVD), and neurodevelopmental disorders in offspring. ***Methods***: We conducted a systematic review of the literature utilizing PubMed for studies published between January 2007 and March 2022. Our search included human cohorts and case control studies following offspring exposed at least to two different types of maternal diabetes clearly identified during pregnancy. Collected outcomes included prevalence, incidence, odds ratio, hazard ratio and risk ratio. ***Results***: Among 3579 published studies, 19 cohorts were eligible for inclusion in our review. The risks for overweight, obesity, type 2 diabetes (T2D), glucose intolerance, metabolic syndrome, and CVD were increased for all types of maternal diabetes during pregnancy. The risk of overweight or obesity in infancy and in young adults was similar between gestational diabetes mellitus (GDM) and type 1 diabetes (T1D). The risk for T2D or abnormal glucose tolerance was double for offspring from GDM mothers compared to offspring from T1D mothers. In contrast, the risk for T1D in offspring at any age until young adulthood was increased when mothers had T1D compared to GDM and T2D. The risk for CVD was similar for all types of maternal diabetes, but more significant results were seen in the occurrence of heart failure and hypertension among offspring from T2D mothers. The risk of autism spectrum disorders and attention deficit/hyperactivity disorders was mainly increased after in utero exposure to preexisting T1D, followed by T2D. ***Conclusions***: Offspring of diabetic mothers are at increased risk for multiple adverse outcomes with the highest risk detected among offspring from T2D mothers. Future work warrants large multiethnic prospective cohort studies that aim to identify the risks associated with each type of maternal diabetes separately.

## 1. Introduction

Environmental and nutritional exposures during prenatal development, as well as other critical developmental periods, can result in an increased risk for non-communicable chronic diseases in adulthood. This concept is known as developmental programming [1]. Based on historical cohorts, the epidemiologist David Barker demonstrated in the early 1990s an association between low birth weight and cardio-metabolic diseases later in life and set the basis for this concept, which is called the “Barker hypothesis” [2]. He hypothesized that alterations of the intrauterine environment induce compensatory responses in tissue structure and function that may persist later in life. These responses may provide an initial benefit to the organism but may be harmful later in life [3]. Initial observations in this field focused on the deleterious effect of low birthweight and undernutrition during fetal life on the risk of cardiovascular diseases (CVD), diabetes and obesity. Subsequent works demonstrated that overnutrition during critical periods of development also contributes to the increased risk of adverse outcomes later in life [4]. Apart from maternal obesity and excessive gestational weight gain, maternal diabetes is a contributor to fetal overnutrition. Early observations on the consequences of maternal diabetes on offspring health come from studies of discordant siblings in the Pima population. Comparing siblings before and after diagnosis of maternal type 2 diabetes (T2D), these studies demonstrated that prenatal exposure to diabetes results in a higher risk of obesity, hyperglycemia and hypertension during the life course of the offspring [5]. Since these studies, many others have reported that these effects are not limited to T2D; they are also observed in the offspring of mothers with type 1 diabetes (T1D) and gestational diabetes mellitus (GDM) [6,7]. The molecular mechanisms through which intrauterine exposure to hyperglycemia would lead to a higher risk of obesity or diabetes are still not well understood. Epigenetic modifications that influence gene expression are among mechanisms that may allow for short-term adaptation of the organism to adverse events during the developmental period [8,9].

Since the pathophysiology and the critical windows of exposure during development are different for each type of maternal diabetes (T1D vs. T2D vs. GDM), the molecular modifications are likely to be different, and subsequently, the level of risk of each adverse outcome later in life.

To date, several meta-analyses have investigated the risk in the offspring for obesity, diabetes, CVD and neurodevelopmental disorders according to the type of maternal diabetes. However, the risk associated with each type of diabetes in pregnancy was assessed separately in comparison to a control population of normoglycemic pregnancies. We propose a round-up of the latest cohort studies that have compared the outcomes of offspring in the same population according to the exposure to different types of diabetes during pregnancy. Separating these different types of diabetes will provide important data to help inform future clinical care of pregnant women with these diseases.

This review aims to compare and synthetize, through a systematic review of the literature based on human data, the level of risk for obesity, glucose intolerance, metabolic syndrome, CVD, and neurodevelopmental outcomes in offspring according to the type of maternal diabetes during pregnancy.

## 2. Methods and Materials:

### 2.1. Literature Search and Data Extraction

This systematic review followed the recommended PRISMA guidelines.

A comprehensive literature search was conducted in PubMed (www.ncbi.nlm.nih.gov (accessed on 1 March 2022) for studies published between January 2007 and March 2022, using keywords: (maternal diabetes OR maternal hyperglycemia OR pregestational diabetes OR hyperglycemia in pregnancy) AND (obesity OR adiposity OR overweight OR glucose intolerance OR diabetes OR cardiovascular outcomes OR metabolic syndrome OR neurodevelopmental outcomes OR autism OR attention deficit disorder in offspring). In addition to database searches, we performed a full text review of studies included in meta-analyses investigating the impact of any type of diabetes on the risk in offspring for obesity, diabetes, CVD, and neurodevelopmental outcomes.

The systematic review was limited to human studies. We restricted our search to studies published in English. Review articles were excluded after a search of the reference lists.

### 2.2. Study Selection

After the primary records were retrieved from PubMed, duplicates were removed. Remaining records’ titles and abstracts were screened, and irrelevant studies were excluded. Full texts of studies deemed relevant were obtained and reviewed in detail for eligibility according to the inclusion criteria.

Publications were included in the final analysis if they met the following inclusion criteria: (1)Studies evaluating fetal exposure to at least two types of maternal diabetes, including GDM, pre-pregnancy T1D and T2D. Studies with unclear diabetes status were excluded.(2)At least one of the following offspring outcomes was used as primary or secondary endpoint: obesity, overweight, adiposity, glucose intolerance, diabetes, cardiovascular outcomes, metabolic syndrome, neurodevelopmental outcomes, autism, or attention deficit disorder.(3)Only randomized controlled trials and prospective or retrospective cohort studies were considered.

Studies were excluded according to the following criteria:(1)Studies with unclear diabetes status (i.e., when the term pregestational diabetes mellitus is used without specifying the type of diabetes).(2)Narrative reviews.(3)Systematic reviews and meta-analyses.(4)Studies evaluating pathophysiology rather than clinical outcomes.

### 2.3. Analysis

Statistical measures of outcomes (prevalence, incidence, odds ratio (OR) or risk ratio (RR)) were collected from studies selected for the review. These outcome measures were organized based on the type of outcome that was evaluated, including diabetes, adiposity, cardiovascular and finally neurodevelopmental outcomes. We defined the occurrence of diabetes as any reports of glucose intolerance, hyperglycemia or any type of diabetes mellitus occurring at any age. Adiposity was defined as the presence of obesity or overweight state. Cardiovascular outcomes included the occurrence of any stage of hypertension, dyslipidemia, or heart failure. Neurodevelopmental outcomes included the occurrence of attention deficit/hyperactivity disorders (ADHD), autism spectrum disorders (ASD), or intellectual disorders.

We also explored the association between maternal blood glucose levels during pregnancy and offspring outcomes whenever it was possible. 

## 3. Results 

### 3.1. Flow Chart

Our literature search identified 3579 articles, of which 77 duplicate records were removed. The remaining 3502 records were screened, and 51 of them matched the inclusion criteria. Then, these studies were assessed for eligibility, 12 narrative review papers, 11 systematic reviews and meta-analyses, 7 studies with unclear diabetic status and 2 studies evaluating pathophysiologic basis of disease were excluded. Thus, our study was based on 19 cohort studies (Figure 1). 

### 3.2. Risk of Overweight and Obesity in Offspring of Diabetic Mothers

Eight cohort studies that analyzed the association between maternal diabetes and overweight/obesity in offspring were identified. Four studies were excluded because pregestational diabetes was not clearly identified as T1D or T2D [10,11,12,13].

Data from the four remaining studies in prospective cohorts are presented in Table 1.

Clausen et al. reported the risk for overweight in young adults exposed either to diet-treated GDM or T1D [14]. The risk of being overweight was increased by about two-fold for both types of maternal diabetes, compared to the group of offspring born to mothers with no diabetes. After adjusting for various parameters, including maternal body mass index (BMI), the risk of being overweight was reduced in GDM, but remained unchanged in T1D. In unadjusted analyses of 317 women with GDM who had an oral glucose tolerance test (OGTT) during pregnancy, for each 1 mmol/L increase in maternal fasting blood glucose, the offspring risk of overweight was increased by 51%, and a 1 mmol/L increase in maternal 2 h blood glucose was associated with a 13% increased risk of overweight. These associations were no longer significant after adjusting for confounding factors. For offspring of mothers with T1D (O-T1D), there was no significant association between maternal blood glucose in the first or third trimester and offspring risk of overweight.

The three other studies followed subjects during childhood (up to 6 years or up 11 years).

Boerschman et al. studied the prevalence of overweight status at 2, 8 and 11 years and found that it was significantly increased in offspring of mothers with GDM (O-GDM), compared to O-T1D [15]. The prevalence of overweight status for O-T1D was not different from that of offspring of non-diabetic mothers. Maternal obesity was a strong predictor of overweight at age 11 years in O-GDM (OR 7.0, 95%CI 1.8–27.7). There were no data in this study on maternal blood glucose levels during pregnancy, and therapy of GDM during pregnancy showed inconsistent correlation with overweight risk in offspring.

Pitchika et al. reported an increased risk for overweight at a median age of 5.5 years for offspring born to mothers with GDM, T1D and T2D, with a higher risk for offspring of mothers with T2D (O-T2D) [16]. Risk for obesity was only increased for O-GDM and O-T1D; however, the sample size for O-T2D was small (*n* = 14). There were only two O-T2D with obesity at age 5.5 years, which explains why the risk for obesity was not increased. The associations for O-GDM were attenuated and became non-significant after adjustment for maternal BMI and gestational weight gain. In contrast, for O-T1D, the associations were not affected when adjusted for maternal BMI and gestational weight gain, but they became non-significant after adjusting for birth weight z-scores. For O-T2D, despite the small sample size, the risk for overweight was attenuated but remained significant after adjusting for maternal BMI, gestational weight gain and birth weight z-scores. There were no data in this study on maternal blood glucose levels during pregnancy. However, there was no association between anthropometric outcomes in the offspring at 5.5 years and the type of antidiabetic treatment during pregnancy, whether it was with insulin (*n* = 72), diet changes (*n* = 243), oral therapy alone (*n* = 1), or lack of a treatment (*n* = 24).

The last cohort study analyzed BMI trajectory after exposure to maternal unmedicated GDM, medicated GDM, T1D or T2D, up to an age of 10 years [17]. This study showed that the age for having a BMI more than 1 standard deviation higher than the BMI for the no diabetes group was 7, 4, 2 and 2.5 years for unmedicated O-GDM, medicated O-GDM, O-T1D and O-T2D, respectively. These results were higher when adjusted for prepregnancy BMI and gestational weight gain (Table 1).

### 3.3. Risk of Type 2 Diabetes or Abnormal Glucose Level

Four cohort studies were identified, two retrospective and two prospective. The two prospective studies were based on the same cohort [18,19]. Only the study from Clausen et al. was included in this review since the second one explored the pathophysiology of the changes in offspring metabolism but did not add data about the risk for T2D. The three remaining studies are presented in Table 2.

Clausen et al. reported the risk for T2D or abnormal glucose (impaired fasting glucose or impaired glucose tolerance evaluated by 75-g glucose tolerance test) in young adults born to mothers with GDM or T1D [18]. Compared to women with no diabetes during pregnancy, women with GDM had a higher mean fasting blood glucose (5.2 versus 4.7 mmol/L) and a higher 2 h blood glucose after OGTT (7.8 versus 5.2 mmol/L) (*p* < 0.0001).

The risks were significantly increased for both types of maternal diabetes compared to the control group with no diabetes, and they remained unchanged after adjustment for maternal overweight among other factors. The risk was much higher in the case of maternal GDM (eight-fold increase in O-GDM, four-fold increase in O-T1D). Additional adjustment for offspring overweight further decreased the association but did not change the pattern (OR (95%CI) 6.83 (2.25–20.71) and 3.19 (1.03–9.90) for O-GDM and O-T1D, respectively). In O-T1D, the risk of T2D or abnormal glucose was significantly associated with elevated maternal glucose in late pregnancy (OR 1.41 (95% CI 1.04–1.91)) per mmol when adjusted for maternal family history of diabetes, maternal overweight, and offspring age.

In a study based on a national registry that assessed mortality and morbidity in the offspring of mothers with diabetes compared with a control cohort of offspring of mothers without diabetes, Nielsen et al. explored the cumulative incidence of all types of diabetes recorded from 8 days of life up to 35 years (mean age 21.5 years) in O-GDM, O-T1D and O-T2D [20]. Data from O-GDM and O-T2D were pooled in the analysis. After 20 years of observation, the cumulative incidence of recorded diabetes in offspring of mothers with T1D was significantly increased compared with offspring of mothers without diabetes. Beyond 20 years of age, the incidence was five- to seven-fold higher in all categories of maternal diabetes. This study has grouped the offspring of mothers with GDM and T2D in the analysis and has considered the incidence of all types of diabetes in offspring [20]. There were 22 offspring with diabetes during the follow-up period, 15 had T1D and 7 had T2D. In this study, women with diabetes during pregnancy were identified by the International Classification of Disease Code (ICD), and therefore, maternal levels of glycemia were not recorded.

In a retrospective cohort, Wicklow et al. explored the incidence rate per year for T2D in O-GDM and O-T2D by age 30 years (mean age 17.7 years) [21]. T2D exposure conferred a greater risk to offspring compared with GDM exposure (3.19 vs. 0.80 cases per 1000 person-years, *p* < 0.001). Compared with no diabetes exposure, any diabetes exposure accelerated the time to the development of T2D in offspring by a factor of 0.72 (95%CI, 0.61–0.65) for GDM and 0.47 (95%CI, 0.45–0.51) for T2D. The database-driven design of the study did not allow for the analysis of maternal blood glucose levels during pregnancy.

### 3.4. Risk of Type 1 Diabetes

Five retrospective studies specifically explored the risk for T1D in offspring exposed to maternal diabetes during pregnancy (Table 3). None of these studies reported data on maternal blood glucose levels during pregnancy.

Algert et al. identified risk factors associated with the onset of T1D before 6 years of age in a cohort of 502,040 singleton live births between January 2000 and December 2005. There were 272 infants admitted with a diabetes mellitus diagnosis before the age of 6 years. Onset of T1D before 6 years of age was associated with maternal T1D, but not with GDM or T2D. Maternal T1D was a strong risk factor for onset of T1D in the child (RR 6.30); however, this was based on only five cases. There were no cases among infants whose mothers had T2D. When separate multivariable models were performed for children diagnosed at <3 years and ≥3 years, maternal T1D remained significantly associated with onset of T1D in offspring (RR 95%CI: 6.72, 2.13–21.2 and 4.34, 1.07–17.7, respectively).

In a Swedish nationwide cohort study comprising children between 0 and 18 years of age, Hussen et al. showed that offspring of mothers with T1D had six times higher risk of having T1D compared with offspring of mothers without diabetes regardless of the mother’s country of birth. Offspring of Nordic mothers with GDM or T2D had an almost doubled risk of having T1D, but in offspring of non-Nordic mothers, there was no association between maternal GDM or T2D and the risk of O-T1D. These data were adjusted for first trimester maternal BMI, among others.

Lee et al. performed a nested case-control study on a cohort of all live births in Taiwan between 2000 and 2005, who were followed until the end of 2008. For each case of T1D in offspring, ten age- and sex-matched controls were randomly selected. The authors found a 5.4-fold increase for the risk of T1D in offspring in the case of maternal GDM. The increase in risk was two-fold higher in O-T1D compared to O-GDM, but it was not statistically significant, probably due to the limited number of mothers with T1D.

Goldacre showed that between the ages of nine months and twelve years, children born to mothers with T1D experienced a significantly higher incidence of T1D than those born to mothers with GDM (HR 7.55 95%CI 6.12–9.33 and 1.3 95%cI 1.00–1.68, respectively).

Lindell et al. performed a case-control study on medical registries of 3231 patients diagnosed with T1D aged 0–19 years. All children with diabetes were matched with four control subjects for date of birth, Sweden’s region of birth and sex. Offspring of mothers with GDM had an almost doubled risk of having T1D, and offspring of mothers with T1D had a five-fold increased risk of having T1D. Additional adjustments including BMI in early pregnancy and gestational weight gain slightly decreased the association but did not change the pattern.

### 3.5. Risk of Cardiovascular Disease and Metabolic Syndrome in Offspring from Diabetic Mothers

Our literature search identified five cohort studies, of which two were excluded because pregestational diabetes was not well defined [11,27]. The inclusion of renal disease as an outcome was initially considered for our review; however, we failed to include it since we only found one systematic review evaluating this outcome [28]. Data are presented in Table 4. 

Clausen et al. studied, in a prospective cohort in Denmark, the risk of overweight and metabolic syndrome in adult offspring of women with diet-treated GDM or T1D [14]. Metabolic syndrome was defined according to the International Diabetes Federation 2006 criteria [29]. Its prevalence was significantly higher in O-GDM compared to O-T1D (24.15% and 14%, *p* value < 0.001, respectively). Based on follow-up data in offspring between 18–27 years of age, the risk for metabolic syndrome was about two-fold higher in O-GDM compared to O-T1D. This risk was reduced after adjusting for multiple confounders in the O-GDM group, but it remained significant in both groups with OR 4.12 (95% CI 1.69–10.06) and OR 2.59 (95% CI 1.04–6.45), respectively (Table 4). In the subgroup analysis of 317 women with GDM, it was noted that for each 1 mmol/L increase in maternal fasting glucose and maternal 2 h blood glucose that the offspring risk of metabolic syndrome was increased by 80% and 18%, respectively. This association remained statistically significant after adjustment for multiple confounders. In the case of O-T1D, a borderline significant association was seen between maternal blood glucose in the third trimester and offspring risk of the metabolic syndrome, which was only present in the unadjusted analysis.

Yu et al. evaluated the associations between the exposure to maternal T1D, T2D or GDM and early onset CVD in offspring during their first four decades of life [30]. Outcomes were identified using the international classification of diseases, 8th and 10th revisions (ICD-8 and ICD-10) codes for CVD. The authors investigated the following types of CVD: ischemic heart disease, cerebrovascular disease, stroke, heart failure, atrial fibrillation, hypertensive disease, deep vein thrombosis, pulmonary embolism, along with other types of CVD. These outcomes were not analyzed according to maternal glucose level.

For all types of maternal diabetes, a 40% to 50% increase in the risk for early onset CVD was seen in offspring. After adjustment, this risk was reduced in the GDM group (HR 95% CI 1.19, 1.07–1.32) compared to the two other groups. Early onset of CVD in offspring was of similar magnitude in maternal T1D and T2D. The authors also specifically explored the risk of occurrence of heart failure. In contrast to the risk of CVD, the risk of heart failure was only significantly increased in the T2D group before and after adjustment with HRs, respectively, 2.41 (95% CI 1.08–5.38) and 2.32 (95%CI 1.04–5.19). The risk of hypertension was also statistically significant even after adjustment in the three groups GDM, T1D and T2D, with a higher HR (2.18 95%CI 1.61–2.95) in the O-T2D group (Table 4) [30].

The most recently published cohort evaluating the association between GDM or T2D and the risk of occurrence of CVD and risk factors for CVD including hypertension, dyslipidemia and T2D was conducted in Canada in 2020 [31]. CVD was defined as the occurrence of either cardiac arrest, myocardial infarction, ischemic heart disease or cerebral infarction. Exposure to maternal GDM or pre-existing T2-D were categorized according to the provided database with no blood glucose level reported. In this cohort, offspring were followed to up to 35 years of age. The crude incidence of CVD increased across the follow-up period, and it was highest among offspring exposed to maternal T2D at 1.04 per 1000/yr. Adjusted HRs and propensity scores were calculated for the main confounding variables and showed that exposure to GDM and T2D was associated with a similar increase for the risk of CVD of about 40%, although at the limit of significance for O-T2D. Both offspring exposed to GDM and T2D were at high risk for the occurrence of CVD risk factors with a higher significance for O-T2D (HR 95%CI 3.44, 2.89–4.11).

**Table 4 nutrients-14-03870-t004:** Risk for cardiovascular disease and metabolic syndrome in offspring according to the type of diabetes during pregnancy.

Author, Year	Countries	Type of Study	Types of Diabetes (Sample Size)	Age at Follow Up	Outcomes	Outcomes with Adjustment
**Clausen et al., 2009 [14]**	Denmark	Prospective cohort	GDM (168) T1D (160)	18–27 years	**Risk for metabolic syndrome (OR 95%CI)** GDM: **5.35 (2.31–12.42)** T1D: **2.73 (1.12–6.64)**	**Adjusted risk for metabolic syndrome (OR 95%CI) ^a^** GDM: **4.12 (1.69–10.06)** T1D: **2.59 (1.04–6.45)**
**Yu et al., 2019 [30]**	Denmark	Population based cohort	GDM (26 272) T1D (22 055) T2D (6537)	Birth- 40 years	**Risk for overall CVD (HR 95%CI) ^b^** GDM: **1.51 (1.36–1.67)** T1D: **1.46 (1.34–1.59)** T2D: **1.44 (1.27–1.62)** **Risk for Heart failure (HR 95%CI)** GDM: 1.61 (0.72–3.60) T1D: 0.95 (0.40–2.29) T2D: **2.41 (1.08–5.38)** **Risk for hypertension (HR 95%CI)** GDM: **2.50 (1.79–3.48)** T1D: **1.81 (1.40–2.35)** T2D: **2.29 (1.69–3.10)**	**Fully adjusted (HR 95%CI) ^c^** **Risk for overall CVD** GDM: **1.19 (1.07–1.32)** T1D: **1.31 (1.20–1.43)** T2D: **1.39 (1.23–1.57)** **Risk for Heart failure** GDM: 1.54 (0.68–3.47) T1D: 0.95 (0.39–2.28) T2D: **2.32 (1.04–5.19)** **Risk for hypertension** GDM: **1.77 (1.27–2.48)** T1D: **1.57 (1.22–2.04)** T2D: **2.18 (1.61–2.95)**
**Guillemette et al., 2020 [32]**	Canada	Retrospective cohort	GDM (8210) T2D (3217)	10–35 years		**Adjusted risk for CVD (HR 95%CI) ^d^** GDM: **1.42 (1.12–1.79)** T2D: 1.40 (0.98–2.01). **Adjusted risk for CVD risk factors (HR 95%CI)** GDM: **1.92 (1.75–2.11)** T2D: **3.44 (2.89–4.11)**

^a^ Adjusted: for maternal age at delivery, maternal pregestational BMI, ethnic origin, family occupational social class. ^b^ Only adjusted for offspring age. ^c^ Adjusted for offspring’s age as time scale, and controlled for calendar year, sex, singleton status, parity, maternal smoking, maternal education, maternal cohabitation, maternal residence at birth, maternal history of CVD before childbirth, paternal history of CVD before birth of the child, and maternal age. ^d^ Adjusted after propensity score matching (PSM): maternal age at offspring birth; socioeconomic Factor Index, rural residence; birth weight; small for gestational age; large for gestational age; and preterm birth. GDM: gestational diabetes mellitus, T1D: type 1 diabetes, T2D: type 2 diabetes, MS: metabolic syndrome, CVD: cardiovascular disease, OR: odds ratio, HR: hazard ratio, CVD risk factors: diabetes, hypertension, dyslipidemia. Statistically significant results are in bold (*p* < 0.05).

### 3.6. Risk of Neurodevelopmental Disorders in Offspring from Diabetic Mothers

Seven cohort studies were identified in our search, of which five were included. The remaining two cohorts were excluded because the pregestational diabetic status was not well categorized [33,34]. The population-based cohort by Kong et al. in Finland in 2020 was included; however, data from this study are not illustrated in Table 5 since the incidence of neurodevelopmental disorders was low in comparison to that of psychiatric disorders [35]. In this study, the incidence of intellectual disorders, ASD, ADHD were, respectively, 0.39%, 0.81% and 1.31%. There were statistically significant correlations between maternal diabetes (both GDM and T2D) and offspring having either intellectual disorders or ADHD, which were only observed when maternal BMI was above 35.

Xiang et al. conducted a multiethnic retrospective cohort to study the ASD risk associated with intrauterine exposure to preexisting T2D and GDM. ASD was defined using ICD-9 codes. Offspring were followed for a median of 5.5 years, 2% were exposed to maternal T2D, and 7.8% were exposed to maternal GDM [36]. The unadjusted average annual incidences of ASD in O-T2D and O-GDM were 3.26, and 2.14 per 1000, respectively (*p* < 0.001). The bivariate analysis showed that exposure to maternal T2D was significantly associated with ASD with HR of 1.59 (95%CI 1.29–1.95), this association remained after adjustment. In contrast, the risk of ASD was not statistically significant after exposure to GDM at any age after adjustment. Even in the stratified bivariate analysis, for mothers with GDM diagnosed on laboratory values confirming a plasma glucose level of 200 mg/dL or higher on the glucose challenge test, or at least two OGTT values exceeding thresholds reported by the authors, the risk of offspring’s ASD was not significantly increased. Only offspring exposed to GDM earlier in pregnancy (before 26 weeks of gestation) were at risk for the occurrence of ASD in the bivariable and the multivariable analysis before and after adjustment with HRs, respectively, 1.63 (95%CI 1.35–1.97) and 1.40 (95%CI 1.14–1.72). 

In 2018, Xiang et al. extended their analysis to study the risk of ASD in offspring exposed to maternal T1D as well [37]. This analysis found similar results for the risk of ASD associated with maternal T2D and GDM (Table 5). It showed that offspring exposed to maternal T1D had the highest risk of ASD in both the adjusted and fully adjusted analyses (HR 2.33 (1.29–4.21) and 2.36 (1.36–4.12), respectively) (Table 4).

The association between ADHD and maternal T1D, T2D and GDM was also studied by Xiang et al. in a retrospective birth cohort in 2018 [38]. They followed offspring from diabetic mothers until up to 18 years of age. ADHD cases were identified using ICD-9 codes. ADHD was not associated with GDM before or after adjustment. In contrast, the occurrence of ADHD was associated with the severity of GDM with a significantly higher HR among offspring exposed to GDM requiring antidiabetic medications (HR 95%CI 1.26, 1.14–1.41), but glucose levels were not reported This association was also found among offspring exposed to maternal T1D and T2D. The risk was the highest for O-T1D, but it decreased and was close to that of O-T2D after adjustment (HR 95%CI 1.56, 1.09–2.25 and 1.43, 1.28–1.59, respectively). Overall, ADHD risk was greatest in children exposed to T1D, followed by those exposed to T2D and GDM requiring medical treatment. The correlation between glycosylated hemoglobin (HbA1c) and the risk of ADHD was not studied due to limited available data. To note, the mean level of HbA1c was most elevated in the case of maternal T1D compared to those with T2D or GDM (7.6%(60 mmol/mol) vs. 6.8% (51 mmol/mol) vs. 6.1% (43 mmol/mol), respectively). 

The most recent population-based cohort study conducted in Sweden by Chen et al. examined the association between the exposure to maternal T1D, T2D or GDM, and the risk of neurodevelopmental disorders, including intellectual disorders, ASD, and ADHD in offspring [39]. These diagnoses were determined using ICD-9 and ICD-10 codes. During the follow-up period, 0.88% of offspring were diagnosed with intellectual disorders, 1.93% with ASD and 4.31% with ADHD, with considerable overlap between diagnoses. Maternal diabetes was associated with increased odds of any neurodevelopmental disorders. This association was statistically significant even after adjustment, with T2D being associated with higher odds compared with GDM and T1D with OR (95%CI), respectively, for any intellectual disorders (2.09, 1.53–2.87), any ASD (1.37, 1.03–1.84), and any ADHD (1.43, 1.16–1.77).

## 4. Discussion

### 4.1. Summary of the Results (Table 6)

The risks for overweight/obesity, T2D/glucose intolerance, metabolic syndrome, and CVD were increased for all types of maternal diabetes. 

**Table 6 nutrients-14-03870-t006:** Summary of the main findings.

Outcomes	Risk for the Offspring According to the Type of Maternal Diabetes
**Overweight/obesity**	Increased risk for all types of maternal diabetes No increased risk for O-GDM after adjustment for maternal BMI Limited data for O-T2D
**T2D/abnormal glucose**	Increased risk for all types of diabetes, limited data Risk for O-GDM > O-T1D Risk for O-T2D > O-GDM
**T1D**	Risk mainly increased for O-T1D
**Metabolic syndrome**	Risk X 2 for O-GDM versus O-T1D, limited data Moderate decrease in risk for O-GDM after adjustment for maternal BMI
**Cardiovascular diseases**	Similar increase in risk for all types of maternal diabetes
**Hypertension**	Similar increase in risk for all types of maternal diabetes
**Heart failure**	Risk X 2 only for O-T2D
**Cardiovascular risk factors**	Risk X 1.5 for O-T2D versus O-GDM
**ASD/ADHD**	Risk mainly increased for O-T1D and O-T2D Risk for O-T1D > O-T2D
**Intellectual disorders**	Increased risk for all types of maternal diabetes Risk X 2 for O-T2D versus O-GDM or O-T1D

GDM: gestational diabetes mellitus; T1D: type 1 diabetes; T2D: type 2 diabetes; ASD: autism disorder; ADHD: attention deficit hyperactivity disorder; O-GDM: offspring of mother with gestational diabetes mellitus; O-T1D: offspring of mother with type 1 diabetes; T2D: offspring of mother with type 2 diabetes.

The risk of overweight or obesity in infancy (up to six years of age) and in young adulthood was similar between GDM and maternal T1D. In contrast, the prevalence of overweight was double for O-GDM up to 11 years of age compared to O-T1D. However, in all studies, for GDM, but not for T1D, the association with overweight/obesity decreased when maternal obesity was taken into consideration in the analyses. There are limited data for T2D. Depending on the study, the risk for overweight/obesity associated with maternal T2D was higher or equivalent to the risk observed for T1D. 

It is difficult to compare the level of risk for developing T2D or abnormal glucose in offspring according to maternal diabetes because of the small number of studies and the different types of design. In young adults, the risk or prevalence for T2D or abnormal glucose level were double for O-GDM compared to O-T1D. For O-GDM, but not O-T1D, the risk was moderately decreased when adjusted for maternal BMI. One study found a four-fold higher incidence rate for T2D in O-T2D compared to O-GDM. 

In contrast, the risk for T1D in offspring at any age until young adulthood was mainly increased when mothers had T1D compared to O-GDM and O-T2D. 

Risk for metabolic syndrome was about two-fold higher in O-GDM compared to O-T1D, and moderately decreased for O-GDM, but not for O-T1D, when adjusted for maternal BMI.

Risk for CVD was similar for all types of maternal diabetes, and the same pattern remained after adjusting for confounding factors. In addition, it was found that all types of maternal diabetes expose to the risk of having hypertension in adulthood, and the risk remained about 1.5 to 2-fold higher after adjustment for confounding factors. 

The risk of heart failure was about two-fold increased only in cases of maternal T2D. The risk of cardiovascular risk factors was 1.5-fold higher for O-T2D, compared to O-GDM.

The risk of ASD and ADHD was increased after in utero exposure to preexisting diabetes T1D, followed by T2D. This statically significant risk was not found with GDM, especially in the case of GDM diagnosed after 26 weeks of gestation, in GDM not requiring medical treatment, or in GDM not associated with obesity. Only one study found an association between GDM and the occurrence of any ASD or any ADHD, but this association was stronger in T1D and T2D [39]

Additionally, maternal diabetes was associated with intellectual disorders, with an approximately two-fold increase in the association among offspring exposed to maternal T2D compared to T1D and GDM.

Most of the studies reported here are large retrospective population-based studies. Only few studies reported on the association between maternal blood glucose levels during pregnancy and offspring risks. When reported, higher blood glucose levels after OGTT in GDM and blood glucose level in late pregnancy appeared to be associated with higher risk of adverse outcomes in offspring.

### 4.2. What This Study Adds Compared to Meta-Analyses?

#### 4.2.1. Risk for Overweight and Obesity

Concerning the risk for obesity in offspring of diabetic mothers, two meta-analyses were published between 2007 and 2022, examining offspring BMI z-score in childhood in relation to GDM and maternal T1D. In both meta-analyses, there were small numbers of studies and participants for GDM and T1D included in the analyses and insufficient data to perform a meta-analysis with regard to O-T2D [40,41]. They both concluded that BMI was higher in O-GDM and O-T1D compared to control groups. After adjustment for maternal pre-pregnancy BMI, this association remained in offspring of T1D, but disappeared in those of GDM mothers. The conclusions of our study are consistent with these results. 

#### 4.2.2. Risk for Abnormal Glucose Levels and T2D

Kawasaki et al. performed a meta-analysis for the risk of all types of diabetes or abnormal glucose levels in offspring in relation to GDM and maternal T1D [41]. There was no significant difference in the risk of developing diabetes during childhood between GDM and controls. In contrast, O-T1D had a higher rate of T2D between infancy and early adulthood compared with control. The conclusions of this meta-analysis are limited because the number of studies and participants included was very low.

#### 4.2.3. Risk for T1D in Offspring

Hidayat et al. performed a systematic review and meta-analysis to evaluate the associations between maternal BMI, maternal diabetes mellitus and maternal smoking during pregnancy, and the risk of childhood-onset T1D in offspring [42]. They found that the greatest risk of T1D in the offspring appeared to be conferred by maternal T1D, followed by GDM, and lastly by maternal T2D. Our results are consistent with this study, showing that the risk of T1D for the offspring is predominantly driven by maternal T1D during pregnancy.

#### 4.2.4. Risk for Cardiovascular Diseases and Hypertension

Only two meta-analyses in our study period looked at the risk for CVD, specifically by evaluating the level of blood pressure (BP) in offspring of GDM mothers [43,44]. A first study reported higher systolic BP in children (2–18 years old) born to GDM mothers [43]. The second study found a significantly higher diastolic and systolic BP in O-GDM compared to control offspring, but included a lower number of studies and participants [44].

Nonetheless, these two studies do not allow one to compare the risk for cardiovascular diseases and hypertension for the offspring between the different types of maternal diabetes.

#### 4.2.5. Risk for Neurodevelopmental Disorders: Intellectual Disorders, ASD, and ADHD

The results of the included cohort studies were consistent with several meta-analyses performed approximately in the same period (2014–2021).

The occurrence of neurodevelopmental disorders has been shown to be associated with in utero exposure to any type of diabetes, knowing that the association was stronger for pregestational diabetes (T1D and T2D) compared to GDM. These results were illustrated by two large meta-analyses that evaluated the occurrence of either ADHD [45] or ADHD along with ASD and intellectual disorders [46], showing up to a two-fold increase in the risk of neurodevelopmental disorders with pre-existing maternal diabetes. These results fall close to what was revealed by our review. However, both studies were limited by the significant heterogeneity of the included reports, limiting the significance of their results. Similarly, a meta-analysis looking at the effect of maternal diabetes on the occurrence of ASD that included multiple case control and cohort studies showed around a 50% increase in the risk of ASD when all types of maternal diabetes were pooled [47]. Unfortunately, the authors did not separate the different types of pregestational diabetes.

Conversely, a trend toward an association between GDM and either ASD and ADHD was found in two different meta-analyses [48,49] that included diverse multiethnic populations. While Zhao et al. observed an increased occurrence of ADHD in offspring of mothers with GDM but not with preexisting diabetes, the results were limited by the small sample size and by population bias. Furthermore, Wan et al. showed an association between maternal diabetes and ASD that was stronger for GDM (RR: 1.62; 95% CI:1.36–1.94) without significant heterogeneity in the studied population.

### 4.3. Strengths and Limitations of the Method

We have selected studies evaluating fetal exposure to at least two types of maternal diabetes in order to compare the long-term consequences of exposure to different types of maternal diabetes within the same populations. For each specific outcome, we found a limited number of studies. Most often, studies used different designs and methods to compare the effects of maternal diabetes on offspring health, which makes the comparison between studies difficult. Additionally, the definition of GDM varied across studies. Furthermore, information on treatment and quality of maternal glycemic control during pregnancy is lacking. Only few studies reported on the association between maternal blood glucose levels during pregnancy and offspring risks.

Furthermore, this strategy allowed us to consider adjustments for the various confounding factors. Moreover, many studies included large populations, often bigger than those reported in meta-analyses. Our study provides a synthesis and a new interpretation of data concerning the risk for obesity, T2D and abnormal glucose levels as well as CVD, hypertension and neurodevelopmental disorders in offspring of diabetic mothers. To the best of our knowledge, this is the first study comparing the impact of each type of maternal diabetes on offspring outcomes. The incidence of obesity and diabetes has been steadily increasing over the last few decades. We have limited our research to the last fifteen years to include studies in which the populations had characteristics that more closely matched those of today’s populations. These data may help clinicians provide clearer prognostic information to diabetic mothers and properly follow those high-risk pregnancies.

### 4.4. What Information to Provide to Pregnant Women?

Finally, all types of maternal diabetes increase the risk of health compromise in adulthood. However, studies are scarce and have different designs, which limit the possibility of comparing the risks between the different types of maternal diabetes with a high level of evidence.

We have shown that the risks differ between studied populations, underlining the fact that genetic traits modulate the effects of maternal diabetes on offspring. There are also many confounding factors, mainly socio-economic status along with postnatal exposures that also influence health in adulthood, which are taken into consideration at variable degrees in the studies [50]. Consequently, the level of risk is not the same for each subject individually, and it is therefore difficult to give accurate and customized information to pregnant women. In most of the studies reported here, maternal glucose levels during pregnancy were not recorded, and we could not determine a link between blood glucose levels at different periods of pregnancy and the level of risk for the offspring. Taken together, our results support the notion that the severity of diabetes matters in affecting the child’s growth and development. Concerning GDM, an observational study on the associations of maternal glycemia on offspring obesity found that worsening hyperglycemia in pregnancy was associated with an increased risk of obesity at ages 5–7 years [51].

Therefore, it is important to inform pregnant women of the importance of tight blood glucose control during pregnancy. Women should be properly educated on the risks of hyperglycemia for their offspring, taking into account the role that normal glycemic values have on maternal and fetal complication incidence, aiming to obtain the best compliance with recommendations. They must also be informed early on, following the diagnosis of T2D, T1D or GDM in their pregnancy, of the crucial impact that glycemic control has on the outcome of their pregnancy and on the offspring’s future [52]. 

### 4.5. Are There Potential Identified Mechanisms?

Our results support the hypothesis of Developmental Origins of Health and Disease (DOHaD). It is theorized that the exposure to certain environmental insults in the early post-natal period or during sensitive periods in utero has a significant impact on short- and long-term health outcomes. However, although most observational studies reported an association between diabetes exposure during pregnancy and adverse risks for future health, they do not provide a direct causal effect. 

Animal models may provide evidence for a direct causal role for exposure to maternal diabetes in utero in determining offspring’s long-term health, thus limiting confounding factors. Different animal models have been designed mimicking diabetes in pregnancy. Nonetheless, the mechanisms behind the long-term consequences of in utero exposure to maternal diabetes remain largely unknown [53]. Epigenetic mechanisms are among the most extensively studied. Alterations in DNA methylation, particularly regarding genes involved in metabolic pathways and cardiovascular disease processes, have been reported [54]. In addition, miRNAs involved in central nervous system development have been found to be dysregulated in maternal diabetic pregnancy [55]. Among other mechanisms, high insulin levels increase the production of free radicals and inflammatory cytokines, which leads to an impairment of antioxidant defense systems resulting in oxidative stress. This mechanism may be involved in vascular remodeling thought to be at the base of the pathogenesis of cardiovascular disease [53]. 

Although they have brought strong evidence for the mechanisms, animal models do not allow for one to differentiate the effects according to the type of maternal diabetes. We can speculate that the timing of the effect of maternal hyperglycemia during pregnancy and its severity may have different molecular consequences and therefore may alter organ development in different ways and affect future health in various patterns.

## 5. Conclusions

In the case of maternal diabetes during pregnancy, the risks of the offspring having adverse outcomes considered in this study are the highest for those born to mothers with T2D, except for the risk of having T1D. The risks associated with GDM are attenuated after controlling for maternal BMI and are especially increased in cases of early GDM or GDM requiring drug treatment. In this situation, there possibly is a great proportion of unrecognized T2D before pregnancy. In order to better identify the risks associated with different types of maternal diabetes, large prospective cohort studies from various countries should be conducted to assess maternal diabetes separately and to adjust for the many confounding factors. Finally, a universal definition of GDM should be considered, as well as the concept of early and late GDM to avoid confusion with unrecognized T2D. Future work is critically needed to assess whether levels of glycemic control may play a role in imparting risk of adverse outcomes in offspring.

## Figures and Tables

**Figure 1 nutrients-14-03870-f001:**
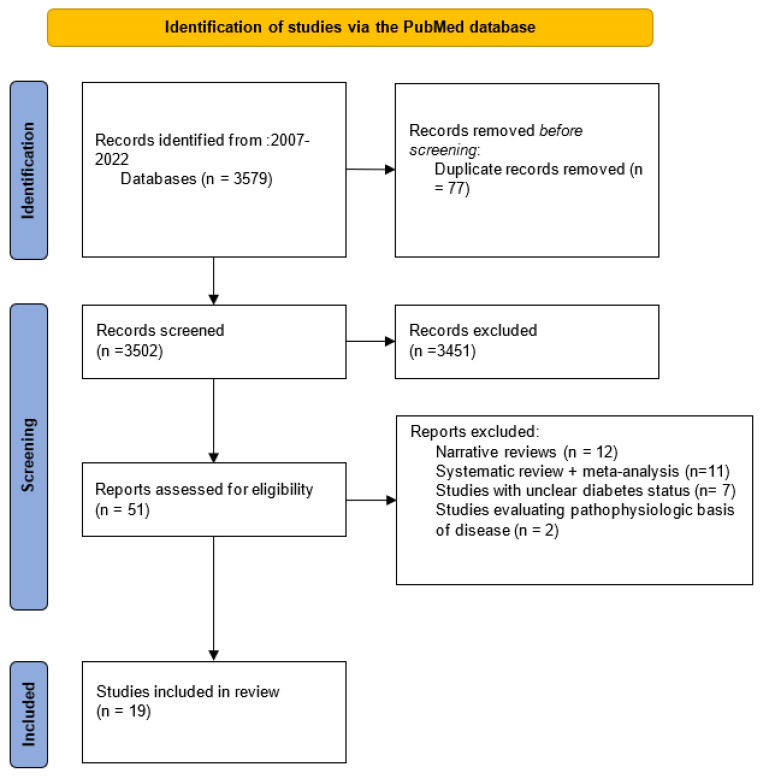
Flow chart.

**Table 1 nutrients-14-03870-t001:** Risk for overweight/obesity according to the type of maternal diabetes during pregnancy.

Author, Years	Countries	Type of Study	Type of Diabetes (Sample Size)	Age at Follow-Up	Outcomes	Outcomes with Adjustments	
**Clausen et al., 2009 [14]**	Denmark	Prospective cohort	GDM (168) T1D (160)	18–27 years	**Risk for overweight (OR 95%CI)** GDM: **2.09 (1.25–3.50)** T1D: **2.15 (1.27–3.62)**	**Fully adjusted (OR 95%CI)**^a^ GDM: **1.79 (1.00–3.24)** T1D: **2.27 (1.30–3.98)**	
**Boerschmann et al., 2010 [15]**	Germany	Prospective cohort	GDM (231) T1D (757)	From 2 years to 11 years	**Prevalence overweight GDM/T1D** 2 years: **17.2%/15.8%** 8 years: **20.2%/11.0%** 11 years: **31.1%/15.8%**	**Prevalence overweight GDM obese/GDM non obese** 2 years: **24.6%/9.2%** 8 years: **36.4%/11.3%** 11 years: **45.8%/11.9%**	
**Pitchika et al., 2018 [16]**	US Germany Finland Sweden	Prospective cohort	GDM (326) T1D (225) T2D (14)	0.25 to 6 years Median age at follow-up: 5.5 years	**Risk for overweight (OR 95%CI) ^b^** GDM: **1.48 (1.14–1.92)** T1D: **1.60 (1.16–2.20)** T2D: **7.39 (2.46–22.23)** **Risk for obesity (OR) ^b^** GDM: **1.98 (1.34–2.93)** T1D: **1.84 (1.09–3.10)** T2D: 2.93 (0.65–13.22)	**Adjusted for maternal pre-pregnancy BMI and GWG (OR 95%CI) ^c^** **Risk for overweight** GDM: 1.14 (0.86–1.51) T1D: **1.50 (1.08–2.09)** T2D: **3.68 (1.14–11.81)** **Risk for obesity** GDM: 1.33 (0.87–2.04) T1D: **1.75 (1.02–3.00)** T2D: 0.94 (0.19–4.60)	**Adjusted for maternal pre-pregnancy BMI, GWG and BW z-score (OR 95%CI) ^d^** **Risk for overweight** GDM: 1.10 (0.82–1.46) T1D: 1.15 (0.81–1.62) T2D: **4.92 (1.40–17.30)** **Risk for obesity** GDM: 1.31 (0.85–2.01) T1D: 1.48 (0.85–2.59) T2D: 1.02 (0.20–5.09)
**Sidell et al., 2021 [17]**	USA (California)	Prospective cohort	Unmedicated GDM (12576) Medicated GDM (6982) T1D (537) T2D (7836)	From birth to 10 years Median age at follow-up: 5 years	**Age (year) with BMI SD > 1 from reference category ^e^** Unmedicated GDM: 7 Medicated GDM: 4 T1D: 2 T2D: 2.5	**Fully adjusted age (year) for maternal pre-pregnancy BMI and GWG** Unmedicated GDM: 9.5 Medicated GDM: 5 T1D: 4.5 TD2: 4.5	

^a^ Fully adjusted for maternal age at delivery, maternal pregestational BMI, offspring age, family occupational social class, maternal hypertension at first visit. ^b^ Adjusted for sex and country. ^c^ Adjusted for sex, country, maternal prepregnancy BMI, breastfeeding, maternal smoking and drinking, gestational weight gain, maternal age, and education. ^d^ Adjusted for sex, country, maternal prepregnancy BMI, breastfeeding, maternal smoking and drinking, gestational weight gain, maternal age, education and birth weight. ^e^ Adjusted for maternal age, race/ethnicity, education, household income, history of comorbidity, gestational age of delivery and child’s sex. GDM: gestational diabetes mellitus; T1D: type 1 diabetes; T2D: type 2 diabetes; OR: odds ratio; BW: birth weight; SD: standard deviation. Statistically significant results are in bold (*p* < 0.05).

**Table 2 nutrients-14-03870-t002:** Risk for type 2 diabetes or abnormal glucose according to the type of maternal diabetes during pregnancy.

Author, Year	Countries	Type of Study	Type of Diabetes (Sample Size)	Age at Follow-Up	Outcomes	Outcomes with Adjustments
**Clausen et al., 2008 [18]**	Denmark Nordic Caucasian	Prospective cohort	GDM (168) T1D (160)	18–27 years	**Risk for abnormal glucose/T2D (OR 95%CI)** GDM: **8.18 (2.83–23.67)** T1D: **3.86 (1.27–11.81)**	**Risk for abnormal glucose/T2D (OR 95%CI) ^a^** GDM: **7.76 (2.58–23.39)** T1D: **4.46 (1.38–14.46)**
**Nielsen et al., 2017 [20]**	Denmark	Retrospective cohort	GDM (136) T1D (521) T2D (34)	8 days–35 years Mean age at follow-up 21.5 years	**Cumulative incidence of all types of diabetes***n* (95%CI) **No diabetes:** <20 years 0.51(0.47–0.55) <35 years 1.14 (1.04–1.26) **T1D: ** <20 years 3.10 (1.62–5.36) <35 years 6.33 (3.97–9.98) **GDM and T2D:** <20 years 1.93 (0.52–5.12) <35 years 4.04 (1.45–8.79)	
**Wicklow et al., 2018 [21]**	Canada	Retrospective cohort	GDM: 4031 T2D: 3788	>7 years up to 30 years Mean age at follow-up 17.7 years	**Incidence rate/1000 persons/years for T2D** GDM: **0.80** T2D: **3.19** **Accelerated failure time * for T2D** 95%CI GDM: 0.72 (0.65–0.61) T2D: 0.47 (0.44–0.57)	

* accelerate failure time: factor by which the time to onset of diabetes is reduced. ^a^ Adjusted for family history of diabetes, maternal overweight, offspring age. GDM: gestational diabetes mellitus; T1D: type 1 diabetes; T2D: type 2 diabetes; OR: odds ratio. Statistically significant results are in bold (*p* < 0.05).

**Table 3 nutrients-14-03870-t003:** Risk for type 1 diabetes in offspring according to the type of maternal diabetes during pregnancy.

Author, Year	Countries	Type of Study	Rate of Type 1 Diabetes (Numbers/‰) or Number of Cases	Length of Follow-Up	Risk for DT1 in Early Childhood	Risk for DT1 in Early Childhood with Adjustments
**Algert et al., 2009 [22]**	Australia New South Wales	Retrospective cohort	272/502,040 0.54‰	Birth to 6 years	**RR (95%CI)** GDM: 1.2 (0.73–1.96) T1D: **6.33 (2.62–15.3)** T2D: 0.0 (0.0–8.4)	
**Hussen et al., 2015 [23]**	Sweden	Retrospective cohort	5771/1,176,155 4.9‰	Birth to 18 years		**Fully adjusted ^a^ (IRR (95%CI))** **Both parents Nordic *** GDM + T2D: **1.85 (1.53–2.23)** T1D: **6.19 (5.30–7.23)** **Both parents non-Nordic** GDM + T2D: 1.13 (0.67–1.92) T1D: **5.80 (2.57–13.08)**
**Lee et al., 2015 [24]**	Taiwan	Retrospective case-control study	Number of T1D: 632 Number of controls: 6320	3–8 years	**OR (95%CI)** GDM: **5.4 (3.62–8.50)** T1D: 10.71 (0.67–171.86) T2D: 0.48 (0.21–1.01)	
**Goldacre, 2018 [25]**	England	Retrospective cohort	2969/3,834,405 7.7‰	Nine months to 12 years Median: 5.7 years	**HR (95%CI)** GDM: 1.3 (1.00–1.68) T1D: **7.55 (6.12–9.33)**	
**Lindell et al., 2018 [26]**	Sweden	Retrospective case-control study	Number of T1D: 3231 Number of controls: 12,948	0–19 years	**OR (95%CI)** GDM: **1.78 (1.07–2.98)** T1D: **5.13 (3.16–8.33)**	**Fully adjusted OR (95%CI) ^b^** GDM: **1.81 (1.08–3.04)** T1D: **4.75 (2.19–7.75)**

* Both parents born in Sweden, Norway, Denmark, Finland or Iceland. ^a^ Adjusted for paternal diabetes, maternal age and BMI, gestational age, maternal smoking, parental level of education, mode of delivery, sex, age at follow-up and birth cohort. ^b^ Adjusted for BMI in early pregnancy, maternal age at delivery, maternal diabetes, parity, smoking during pregnancy, gestational weight gain and pregnancy length. GDM: gestational diabetes mellitus; T1D: type 1 diabetes; T2D: type 2 diabetes; RR: relative risk; IRR: incidence rate ratio; HR: hazard ratio; OR: odds ratio. Statistically significant results are in bold (*p* < 0.05).

**Table 5 nutrients-14-03870-t005:** Risk for neurodevelopmental outcomes including intellectual disorder, autism disorder (ASD) and attention deficit hyperactivity disorder (ADHD). GDM: gestational diabetes mellitus; T1D: type 1 diabetes; T2D: type 2 diabetes; OR: odds ratio, HR: hazard ratio. Significant results are in bold (*p* < 0.05).

Author, Year	Country	Study Type	Type of Diabetes (Sample Size)	Follow Up	Outcomes	Outcomes with Adjustment
**Xiang et al., 2015 [36]**	USA (California)	Retrospective longitudinal cohort	GDM ≤ 26 weeks (7456) GDM > 26 weeks (17 579) T2D (6496)	3–17 years median age at follow-up 5.5 years	**ASD (HR 95%CI)** (Adjusted for birth year) GDM at any age: **1.18 (1.04–1.33)** GDM ≤ 26 weeks: **1.63 (1.35–1.97)** GDM > 26 weeks: 0.98 (0.84–1.15) DT2: **1.59 (1.29–1.95)**	**Fully adjusted ^a^ (HR 95%CI)** GDM at any age: 1.03 (0.90–1.17) GDM ≤ 26 weeks: **1.40 (1.14–1.72)** GDM > 26 weeks: 0.86 (0.73–1.02) T2D: **1.30 (1.04–1.62)**
**Xiang et al., 2018 [37]**	USA (California)	Retrospective longitudinal cohort	GDM ≤ 26 weeks (11 922) GDM > 26 weeks (24 505) T2D (9453) T1D (621)	1–22 years median age at follow-up 6.9 years	**ASD (HR 95%CI) ^b^** GDM ≤ 26 weeks: **1.30 (1.12–1.51)** GDM > 26 weeks: 0.99 (0.88–1.12) T2D: **1.45 (1.24–1.70)** T1D: **2.36 (1.36–4.12)**	**Fully adjusted ^c^ (HR 95%CI)** GDM≤ 26 weeks: **1.26 (1.08–1.47)** GDM > 26 weeks: 0.98 (0.87–1.10) T2D: **1.39 (1.18–1.62)** T1D: **2.33 (1.29–4.21)**
**Xiang et al., 2018 [38]**	USA (California)	Retrospective birth cohort	GDM (29 534) T2D (7822) T1D (522)	4–18 years	**ADHD (HR 95%CI)** (Adjusted for birth year and random siblings) GDM: 0.94 (0.88–1.00) **T2D: 1.40 (1.26–1.56) T1D: 1.97 (1.39–2.79)**	**Fully adjusted ^d^ (HR 95%CI)** GDM: 1.02 (0.96–1.09) **T2D: 1.43 (1.28–1.59)** **T1D: 1.56 (1.09 –2.25)**
**Chen et al., 2021 [39]**	Sweden	Retrospective cohort	GDM (21 325) T1D (17 444) T2D (1679)	6–29 years	**OR 95%CI** (Adjusted for birth year and sex) **Any intellectual disorder:** GDM: **1.68 (1.50–1.89)** T2D: **3.57 (2.59–4.92)** T1D: **1.91 (1.69–2.16)** **Any ADHD:** GDM: **1.17 (1.10–1.25)** **T2D: 1.78 (1.44–2.19)** T1D: **1.42 (1.33–1.53)** **Any ASD:** GDM: **1.43 (1.32–1.56)** T2D: **1.85 (1.39–2.48)** T1D: **1.48 (1.35–1.63)**	**Fully adjusted ^e^ (OR 95%CI)** **Any intellectual disorder:** GDM: **1.30 (1.15–1.46)** T2D: **2.09 (1.53–2.87)** T1D: **1.58 (1.40–1.79)** **Any ADHD:** GDM: **1.16 (1.08–1.23)** **T2D: 1.43 (1.16–1.77)** T1D: **1.21 (1.13–1.29)** **Any ASD:** GDM: **1.30 (1.20–1.42)** T2D: **1.37 (1.03–1.84)** T1D: **1.29 (1.17–1.42)**

^a^ Adjusted for maternal age, parity, education, household income, race/ethnicity, history of comorbidity, and sex of the child, and gestational age and birth weight. ^b^ Adjusted for birth year, maternal age, parity, education, median household income based on census tract of residence, self-reported race/ethnicity, history of comorbidity (≥1 diagnosis of heart, lung, kidney, or liver disease; cancer); and sex of the child. ^c^ Adjusted for c + smoking and prepregnancy BMI. ^d^ Adjusted for random sibling effect, birth year, maternal age at delivery, parity, education, household income, maternal race/ethnicity, history of comorbidity, history of maternal ADHD, and sex. ^e^ Adjusted for parental educational level, income, immigration status and history of inpatient psychiatric care; children’s birthplace in Sweden; and maternal age, BMI, parity, smoking during pregnancy and PCOS. GDM: gestational diabetes mellitus; T1D: type 1 diabetes; T2D: type 2 diabetes; OR: odds ratio, HR: hazard ratio. Significant results are in bold (*p* < 0.05).

## Data Availability

Data sharing not applicable. No new data were created or analyzed in this study. Data sharing is not applicable to this article.

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
