# Peer review of "How Do the Different Types of Maternal Diabetes during Pregnancy Influence Offspring Outcomes?"

_nutrients, 2022, doi:10.3390/nu14183870_

Round 1

Reviewer 1 Report

This systematic review provides insight into the impact of different types of maternal diabetes on the risk for obesity, glucose intolerance, metabolic syndrome, CVD, and neurodevelopmental outcomes in the offspring.

The methodology is rigorous and followed the PRISMA checklist. The search methods are adequately described, although I wonder why the authors decided to exclude articles published before 2007. Please provide a clear explanation.

The results are overall clearly presented and tables are adequately organized according to the different offspring outcomes.

I found table 6 very useful to summarize the large amount of data presented in this review.

Strengths and limitations are not clearly stated. It seems that the author reported the objective of the study as a strength. Please try to make it clearer and rephrase the paragraph.

Overall the paper is valuable and adds knowledge on the impact that each type of diabetes has on offspring's long term outcomes, helping clinicians to provide clearer prognostic information to diabetic mothers and properly follow those high-risk pregnancies.

Reviewer 2 Report

I congratulate the authors preparing a sound and interesting manuscript. I have only minor remarks:

1. Lines 122-123: "Adiposity was defined as the presence of outcomes of obesity...". I think that the word "outcomes" is not necessary.

2. I think that such abbreviations as O-T1D or O-GDM should be better clarified as they are confusing when appear for the first time in the text.

3. Page 8: Expression "in young adults exposed either to GDM or T1D" is unclear and may be misleading. The exposition concerns their intrauterine period of life, I suppose. 
